# Protective Performance of Zn-Al-Mg-TiO$_2$ Coating Prepared by Cold Spraying on Marine Steel Equipment

**Kunkun Wang [1], Shouren Wang [1,\*], Tianying Xiong [2], Daosheng Wen [1], Gaoqi Wang [1], Wentao Liu [1] and Hao Du [2]**

[1] School of Mechanical Engineering, University of Jinan, Jinan 250022, China; 20172120531@mail.ujn.edu.cn (K.W.); me_wends@ujn.edu.cn (D.W.); me_wanggq@ujn.edu.cn (G.W.); me_liuwt@ujn.edu.cn (W.L.)

[2] Institute of Metal Research, Chinese Academy of Sciences, Shenyang 110016, China; tyxiong@imr.ac.cn (T.X.); hdu@imr.ac.cn (H.D.)

\* Correspondence: me_wangsr@ujn.edu.cn

**Abstract:** According to research, we have learned that zinc has excellent cathodic protection performance, that the corrosion products of aluminum and magnesium can form dense and stable passivation films to protect internal materials of coatings, and that TiO$_2$ has excellent photocatalytic self-cleaning performance which will form a physical adsorption film on the surface to isolate the external corrosion solution. In this paper, a Zn-Al-Mg-TiO$_2$ pseudo alloy coating was prepared by cold spray technique on a Q235 substrate. The protective performance of Zn-Al-Mg-TiO$_2$ for marine metal equipment was studied using dynamic salt water corrosion testing, electrochemical testing, and friction and wear testing. The microstructure, composition, and wear marks of coatings were observed using a scanning electron microscope (SEM), energy dispersive spectrometer (EDS), and white-light interferometer. The results show that the Zn-Al-Mg-TiO$_2$ coating has excellent corrosion and wear resistance, which can provide long-term and stable protection for the substrate.

**Keywords:** Zn-Al-Mg-TiO$_2$ coating; marine anticorrosion; corrosion; friction and wear

## 1. Introduction

Ships and drilling platforms are important tools for humans to develop to utilize marine resources, but a large amount of halogen ions in seawater will cause serious corrosion to marine steel equipment, which greatly reduces the service life of equipment. Because of this, scientists have carried out a lot of research to protect equipment from corrosion [1,2].

In the field of anticorrosive coatings for marine equipment, Zn-Al series coatings prepared by thermal spraying and hot-dip plating are the focuses of current research [3–5]. As early as the 1920s, research on Zn-Al protective coating was carried out abroad and applied in practice [6]. It was found that Zn-Al alloy coatings combine the advantages of Zn coatings and Al coatings. The Zn-Al coating not only has excellent cathodic protection but also a passivation film formed on the surface by the corrosion products of Al, which can effectively slow down the corrosion rate [7,8]. In recent years, people have tried to add Mg and TiO$_2$ to Zn-Al coatings for research. The results show that the addition of Mg can significantly improve the corrosion resistance of the coatings [9–11]. Caizhen et al. studied a Zn-Al-Mg coating prepared by hot dipping and found that the addition of magnesium can not only improve the microhardness of the coating but can also form special corrosion products which can block the passage of oxygen and water and reduce the corrosion rate [12]. Joung et al. studied Zn-Mg coatings and found that the structure of the corrosion product film changes and that the corrosion

resistance of the coating is significantly improved [13]. In addition, $TiO_2$ is a marine antifouling and self-cleaning material with excellent chemical stability [14,15]. Under ultraviolet radiation, $TiO_2$ reacts with water and oxygen to form hydroxyl groups with the contact angle of the liquid with the surface of the coating being less than 5°. A physical adsorption film forms on the surface and improves the corrosion resistance of coatings [16–19].

After practical application, it has been found that there are some problems with coatings prepared by thermal spraying and hot-dip plating. The material becomes semi-molten or molten during thermal spraying and hot-dip plating, and the coating is oxidized, generating thermal stress and porosity at high temperature. This seriously affects the corrosion resistance of the coating and shortens the protection period. Cold spray is a surface spraying technology. It has been widely used in recent years. Compared with thermal spraying and hot dip plating, cold spraying has the following advantages: (1) in the working process, the working temperature of cold spraying is low and the material does not need to be heated, meaning cold spray is suitable for the preparation of heat-sensitive material coatings; (2) the velocity of the airflow accelerated by the laval nozzle reaches a supersonic level; (3) material particles are bombarded to the surface of the substrate by high velocity airflow, and the material particles undergo serious plastic deformation, which can improve the structure density and bonding strength of the coating [20–22]. Therefore, in this paper, a $Zn$-$Al$-$Mg$-$TiO_2$ coating was prepared by cold spraying on a Q235 substrate. The protective performance of this $Zn$-$Al$-$Mg$-$TiO_2$ coating for marine metal equipment was studied by dynamic salt water corrosion testing, electrochemical testing, and friction and wear testing. The microstructure, composition, and wear marks of coatings were observed using a scanning electron microscope (SEM), an energy dispersive spectrometer (EDS) and a white-light interferometer. In this work we discuss the corrosion and wear resistance of the $Zn$-$Al$-$Mg$-$TiO_2$ coating to provide some reference for further research on marine metal anti-corrosion coating.

## 2. Experimental Methods

### 2.1. Preparation of Coatings

$Zn$-$Al$-$Mg$-$TiO_2$ coatings were prepared on a $200 \times 30 \times 3$ mm$^3$ Q235 substrate (a common carbon structural steel widely used in ships, drilling platforms, and other offshore buildings, etc.) by cold spraying (DyMET423, Beijing Chuangxuan Bide Tech&Trade Ltd., Beijing, China), with the mass fraction of $Zn$, $Al$, $Mg$, and $TiO_2$ powder being 13:3:1:3. The substrate was blasted with steel grit and degreased in alcohol ultrasonically before spraying. The working parameters of cold spraying were as follows: (1) the working gas was ordinary compressed air with a pressure of 2 MPa and the temperature was 300 °C; (2) the distance between the spray nozzle and the substrate was 20 mm; (3) the moving speed of the spray nozzle was 2 mm/s.

The steel plate was cut into rectangular samples with dimensions of $10 \times 10 \times 3$ mm$^3$ by wire electrical discharge machining (DK774, Taizhou Chuangyuan Machine Tool Co., Ltd., Taizhou, Jiangsu, China) and gently abraded with 800, 1000, 1500, 2000, and 2500 mesh sandpaper, after which the sample was decontaminated and degreased by ultrasonic cleaner (JP-040s, Shenzhen Jiemen Cleaning Equipment Ltd., Shenzhen, China).

### 2.2. Testing and Analysis of Coatings

In this paper, the corrosion and wear performance resistance of a $Zn$-$Al$-$Mg$-$TiO_2$ coating were studied via immersion testing in dynamic salt water, wear testing, and electrochemical testing. In order to reflect the universality of the experiment, the experiments were repeated twice, taking the most representative group for the study.

In order to simulate the marine environment, the magnetic stirrer was used to stir the corrosive medium for the dynamic salt water corrosion test. The test equipment consisted of a magnetic stirrer (cjj79-1, Changzhou Jintan Chenyang Electronic Instrument Factory, Changzhou, Jiangsu, China) and a 1 L beaker. The speed of the magnetic stirrer was set to 1000 r/min. Before the test, the rest of the

sample surfaces were sealed by silicone rubber resin. Four samples were corroded in solution for 144, 240, 480, and 720 h respectively. The corrosive medium was 3.5% NaCl solution.

Reciprocate friction wear testing was carried out under non-lubricated conditions using an RTEC friction and wear tester (MFT-50, San Jose, CA, USA). The test period was 20 min and the Q235 friction pair moved at a reciprocating frequency of 4 Hz. After the test, the wear mass of the sample was measured with an electronic scale and a white-light interferometer (USP-Sigma, Saint Louis, MO, USA) was used to observe the wear scarring of the coating. In order to avoid accidentality, this test was carried out three times for each sample and the test loading forces were 20, 25, and 30 N, respectively.

Electrochemical testing was carried out using an electrochemical workstation (CHI604E, Shanghai Chenhua Instruments Co., Ltd., Shanghai, China). In order to study the effects of the physical adsorption film formed by $TiO_2$ on the corrosion resistance of coating, the test was divided into two groups: those under light and those under non-light conditions. An ultraviolet lamp (20 W) which was perpendicular to the sample was used as the light source in first group; in the second group of tests, the beaker was covered with a blackout cloth. The experiment used the traditional three-electrode mode, with the counter electrode being a platinum electrode, the reference electrode being a calomel electrode (standard potential: 0.2801 V) and the working electrode being a $10 \times 10 \times 3$ mm$^3$ coating sample. Before the test, the rest of the sample surfaces were sealed by silicone rubber resin. Then, the open circuit potential and potentiodynamic polarization curves of the sample were tested by being immersed in static 3.5% NaCl solution for different time periods. The test time of the open circuit potential was kept above 1 h until the potential fluctuation range was less than $1 \times 10^{-3}$ V, and the test interval was 24 h. In addition, the initial potential, final potential, and scan rate of the potentiodynamic polarization curves plot were −1.8 V, −0.8 V, and 10 mV/s, respectively, and the test interval was 48 h.

A scanning electron microscope (SEM, JSM-7610F, JEOL, Tokyo, Japan) was used to observe the microscopic morphology and corrosion product of the coating using secondary electron imaging, with the accelerating voltage being 10 kV. The distribution of elements was measured using an energy dispersive spectrometer (EDS, JSM-7610F, JEOL).

## 3. Results and Discussion

### 3.1. Microstructure of the Coating

The surface and cross-sectional micro-morphology of the Zn-Al-Mg-TiO$_2$ coating are shown in Figure 1. Figure 1a shows the surface morphology, which has the typical characteristics of coatings prepared by cold spraying, with plastically deformed material particles shown which are distributed in groups, these groups mainly including bright and dark forms. Overall, these groups are uniformly distributed on the coating surface, and there are no obvious holes. As shown in Figure 1b, the upper layer is the coating and the lower layer is the substrate, and a part of the coating surface appears on the top of the figure, because the cross section is not completely perpendicular to the lens. In the same way as the surface micro-morphology, material particles of the cross section are distributed in groups, but the material particles' structure of the coating cross section has a flat shape and there are no obvious holes in the particle joint zone that indicate the material particles have undergone severe plastic deformation; in addition, the structure of the coating is dense.

The mapping spectrogram of the morphology shown in Figure 1a was prepared using an energy dispersive spectrometer (EDS) to study the distribution of the components and the elemental composition of the different morphologies, as shown in Figure 2. The Zn element accounts for the largest proportion and is distributed in groups. The content of Al, Ti, and O is relatively rich and is distributed around the Zn-rich phase. The distribution range of Ti and O are basically the same because they are sprayed on the substrate in the form of titanium dioxide particles. The content of Mg is the least and is distributed in groups. By comparing with the micro-morphology of Figure 1a, it can be seen that the distribution of Zn elements and the bright groups of Figure 1a are the same, and that the Al, Mg, Ti, and O elements being distributed in the dark area of Figure 1a indicates that the

bright morphology is the Zn-rich phase and the dark morphology is composed of aluminum, titanium dioxide, and magnesium.

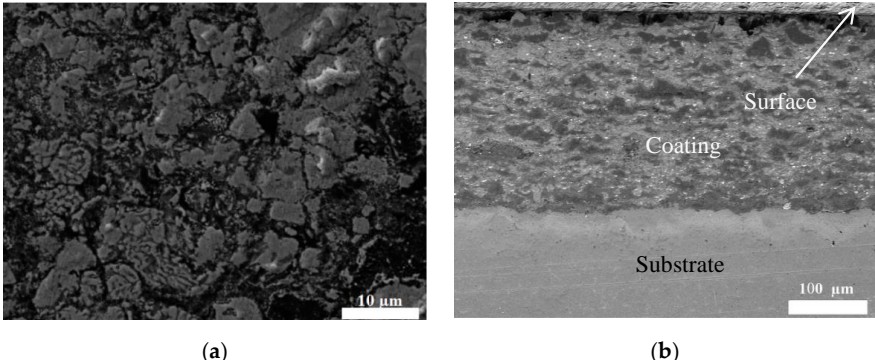

(**a**)  (**b**)

**Figure 1.** The surface and cross-sectional micro-morphology of the Zn-Al-Mg-TiO$_2$ composite coating: (**a**) surface micro-morphology; (**b**) cross-sectional micro-morphology.

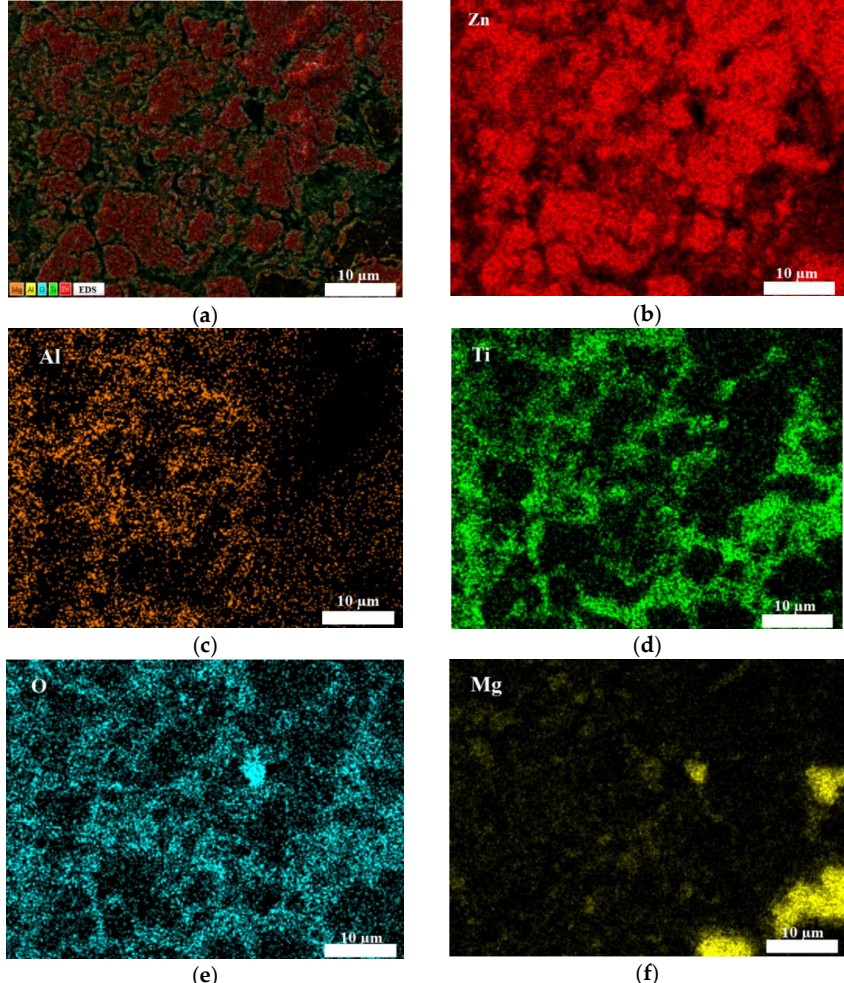

**Figure 2.** Energy dispersive spectrometer (EDS) mapping spectrogram of Figure 1a: (**a**) integrated mapping spectrogram; (**b**) Zn element mapping spectrogram; (**c**) Al element mapping spectrogram; (**d**) Ti element mapping spectrogram; (**e**) O element mapping spectrogram; (**f**) Mg element mapping spectrogram.

### 3.2. Analysis of the Results of Dynamic Salt Water Corrosion Testing

Figure 3 shows surface macroscopic morphology images of the four Zn-Al-Mg-TiO$_2$ coating samples which were immersed in dynamic salt water for 144, 240, 480, and 720 h, respectively. At the initial stage of the immersion, the surface color of coating is darkened and is without obvious pits. Then, the surface of the coating is covered with a thinner white corrosive product after being immersed for 480 h. As the corrosion time goes on, the thickness of the white corrosion products on the coating surface increases gradually. In order to study the morphology and the distribution of corrosion products, the surface micro-morphology of the four Zn-Al-Mg-TiO$_2$ coating samples are magnified 1000×, as shown in Figure 4. At the initial stage of the immersion, there are two kinds of corrosion morphologies on the surface. The upper layer is an agglomerate morphology with a loose structure and the lower layer is a lamellar morphology with a compact structure (in Figure 4a). It is said that the lamellar morphology is the typical corrosion morphology of zinc [23]. Hence, the formation of a large number of lamellar corrosion products of Zn is the reason for the surface color of the coating being darkened at the initial stage. However, as shown in Figure 4b, after having been corroded for 288 h, the agglomerate morphology on the surface of the coating gradually increases, and there are a lot of holes which appear between the lamellar corrosion products, which decreases the structural compactness of the lamellar corrosion products and provides access for the corrosive medium. After 480 h, the surface of the coating is covered with a large amount of agglomerate morphologies, which block the corrosion holes. With the passage of corrosion time, the amount of agglomerate morphology on the surface gradually increases and the structure of them on the surface of coating becomes denser, providing a protective film for the coating surface, which can prevent the intrusion of corrosive solutions and improve the protection period of the coating.

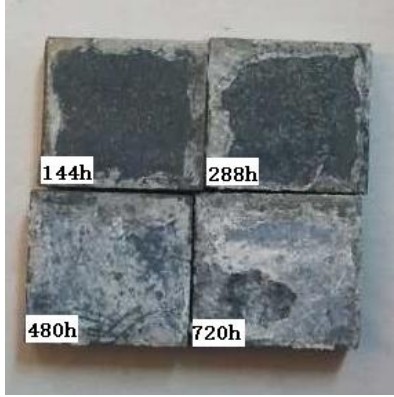

**Figure 3.** The surface macro-morphology of the four Zn-Al-Mg-TiO$_2$ coating samples immersed in dynamic salt water for 144, 240, 480, and 720 h respectively.

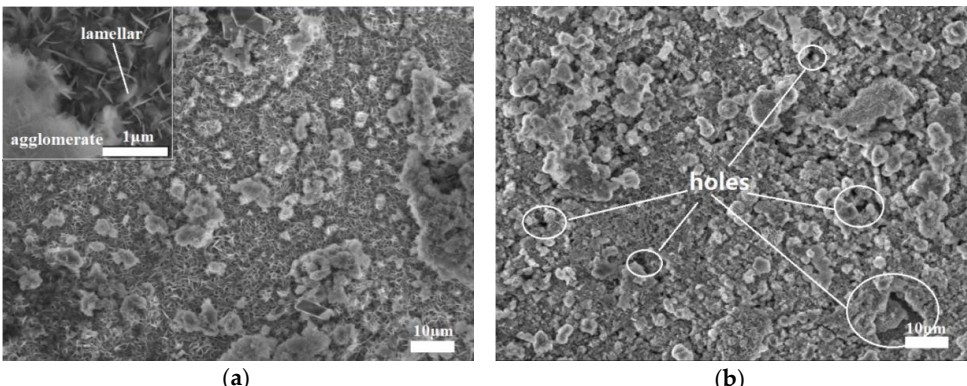

(**a**)                                        (**b**)

**Figure 4.** *Cont.*

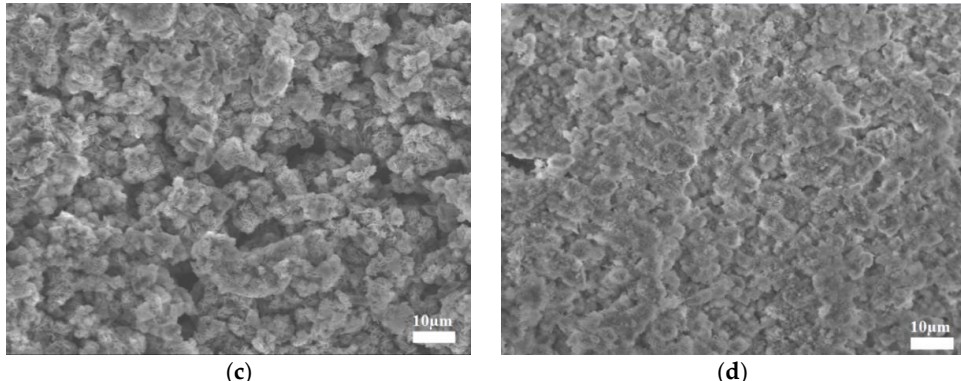

(**c**)                    (**d**)

**Figure 4.** The surface micro-morphology of the four Zn-Al-Mg-TiO$_2$ coating samples, magnified 1000×, after experiencing different corrosion times: (**a**) corrosion 144 h; (**b**) corrosion 240 h; (**c**) corrosion 480 h; (**d**) corrosion 720 h.

Figure 5 shows the EDS line scan image of the sample immersed in dynamic salt water for 288 h. From Figure 5b, we can see that there is a large amount of oxygen and chlorine, which means corrosion produces a lot of oxides and chlorides. Within the scan range, the content of aluminum, magnesium, and titanium is less, and the fluctuation range is small. However, the content of zinc, oxygen, and chlorine is richer, and is mainly concentrated on the lamellar. This phenomenon indicates that the content of aluminum, magnesium, and titanium dioxide is less and is not corroded, but that zinc is corroded seriously, and that the lamellar morphology is mainly due to the corrosion products of zinc, chlorine and oxygen. The metallic element in the dark region is mainly zinc, indicating that the region is a zinc-rich phase, but the content of zinc is significantly lower than in the other Zn-rich phases, which indicates the dark region represents holes or cracks.

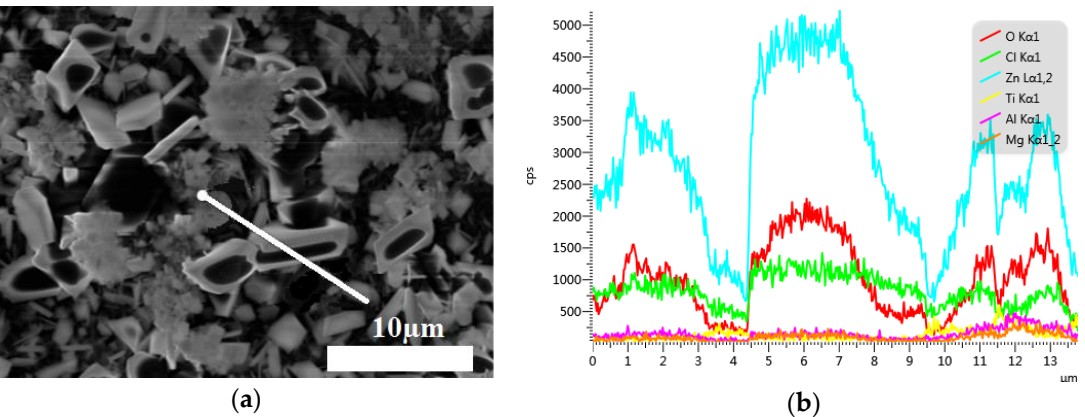

(**a**)                    (**b**)

**Figure 5.** EDS line scan image of a sample immersed in dynamic salt water for 288 h.

*3.3. Analysis of the Results of Electrochemical Testing*

Figure 6 shows the time-open circuit potential chart of the Zn-Al-Mg-TiO$_2$ coating in 3.5% NaCl solution under ultraviolet and no ultraviolet conditions, with the reference electrode being a calomel electrode. The open circuit potential of the test under the two conditions indicates the same trend. At the initial stage of corrosion, the open circuit potential of each is about −1 V, after which they begin to drop sharply, with the open circuit potential under ultraviolet conditions beginning to decrease slowly after falling to −1.1847 V. When the open circuit potential drops to a certain range, it starts to increase slowly.

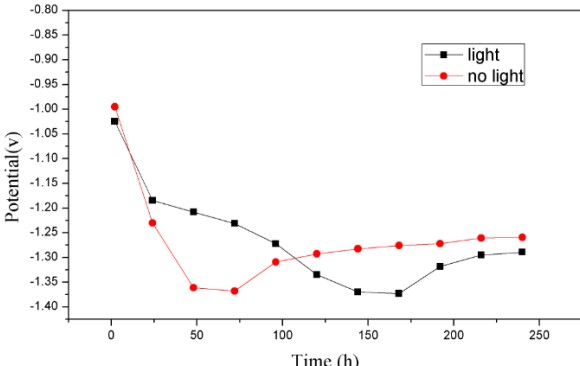

**Figure 6.** Time-open circuit potential chart of the Zn-Al-Mg-TiO$_2$ coating in 3.5% NaCl solution under ultraviolet and no ultraviolet conditions (reference electrode: calomel).

The main reason for this phenomenon is that a large number of zinc groups are corroded at the initial stage of immersion, so corrosion cracks and holes are formed on the surface of the coating and the corrosive medium enters the inside of the coating. The amount of the Zn-rich phase exposed to the corrosive medium increases greatly, meaning the open circuit potential of the coating decreases rapidly. After a period of ultraviolet radiation, the photocatalytic reaction of TiO$_2$ causes the surface of the coating to be super-hydrophilic and the water film to be attached to the surface of coating, separating the coating from the corrosive medium. Hence, the open-circuit potential of the coating under ultraviolet conditions begins to decline slowly. At the later stages of immersion, the active metal on the surface of the coating is covered by a dense and stable passivation film. As the thickness of the passivation film increases, the open circuit potential moves in the positive direction.

Figure 7 shows the potentiodynamic polarization curves of the Zn-Al-Mg-TiO$_2$ coating after immersion in 3.5% NaCl solution for 48, 96, 144, 192, and 240 h respectively; for these tests a calomel electrode was used. As shown in Figure 7, the polarization potential of the substrate is −0.973 V and the polarization potential of the two sets of tests are lower than the substrate. This indicates that the coating is a cathode, providing cathodic protection for the substrate during the corrosion process. Furthermore, the potentiodynamic polarization curves of the two sets of tests at different corrosion times indicate that when the potential rises to a certain value, the corrosion current density does not change with the increasing potential, and the corrosion rate is completely controlled by diffusion processes, which means passivation behavior occurs on the surface of the coating and the corrosion current density of the coating is equal to limiting current density. As shown in Figure 7b, for the no-ultraviolet test, the corrosion current density increases first and then decreases, indicating that the corrosion rate of the coating increases gradually at the beginning of corrosion. As the immersion time increases, and as the corrosion progresses to a certain extent, the corrosion rate stops increasing and begins to gradually decrease. This phenomenon proves the above judgment. In the initial stage of corrosion, a lot of the active metals of the coating are corroded, accelerating the corrosion rate. As a large amount of corrosion products accumulate on the surface of the coating, the active metals are covered by corrosion products and the corrosion rate gradually decreases. With this trend in mind, and comparing the limiting current density of the two sets of experiments, we can find that the corrosion current density of the ultraviolet radiation experiment is always lower than that of the no ultraviolet radiation experiment. This indicates that the photocatalytic self-cleaning effect of titanium dioxide can effectively reduce the corrosion rate of the coating.

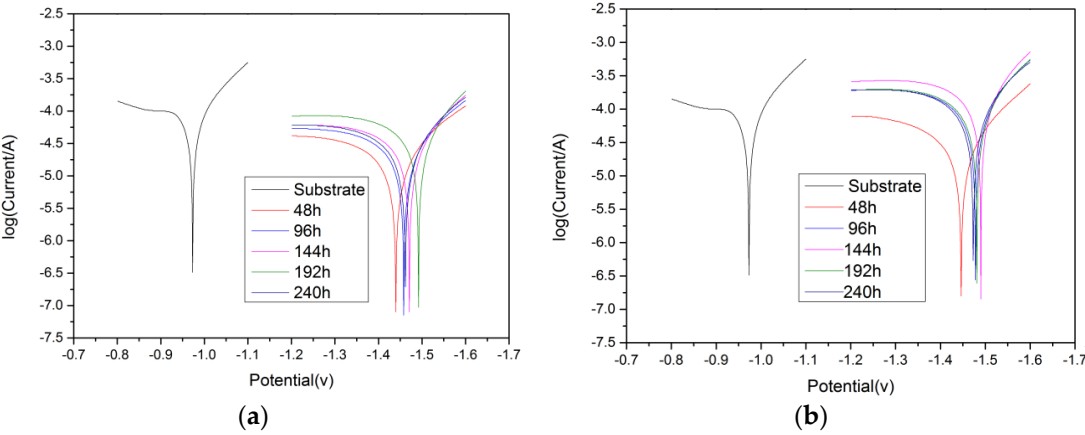

**Figure 7.** Potentiodynamic polarization curves of Zn-Al-Mg-TiO$_2$ coating in 3.5% NaCl solution for different time periods (reference electrode: calomel): (**a**) potentiodynamic polarization curves of the coating under ultraviolet conditions; (**b**) potentiodynamic polarization curves plot of the coating under no ultraviolet conditions.

### 3.4. Analysis of the Wear Resistance of the Coating

Damage to the coatings of marine steel substrates includes corrosion and wear. The NaCl-corrosion and electrochemical tests showed that the Zn-Al-Mg-TiO$_2$ coating has excellent corrosion resistance. In order to study the friction and wear properties of the coating, three sets of friction and wear tests were carried out. The friction coefficient is shown in Figure 8, and Figure 9 indicates the 3D morphology of the wear marks under different forces, which is the partially enlarged view taken from the center of the wear mark. In different loading force tests with the Q235 friction pair, the coating friction coefficient fluctuates between 0.3933–0.3601 and wear masses are below 10 mg. When the loading force is 20 N, the wear mark depth of the coating is 97 μm and the wear mark is relatively smooth, with no bulk material falling off and the wear mass being only 3.1 mg. When the loading force increases, the wear mass and the internal morphology of the wear scarring do not change much, and there are no obvious pits. The results indicate that the Zn-Al-Mg-TiO$_2$ coating has a dense structure, high bonding strength, and great wear resistance.

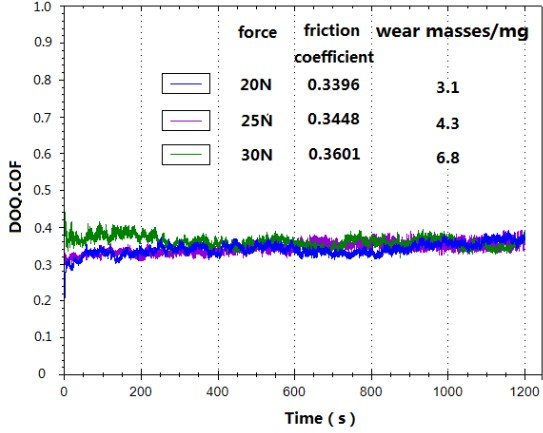

**Figure 8.** The friction coefficient curve of the Zn-Al-Mg-TiO$_2$ coating.

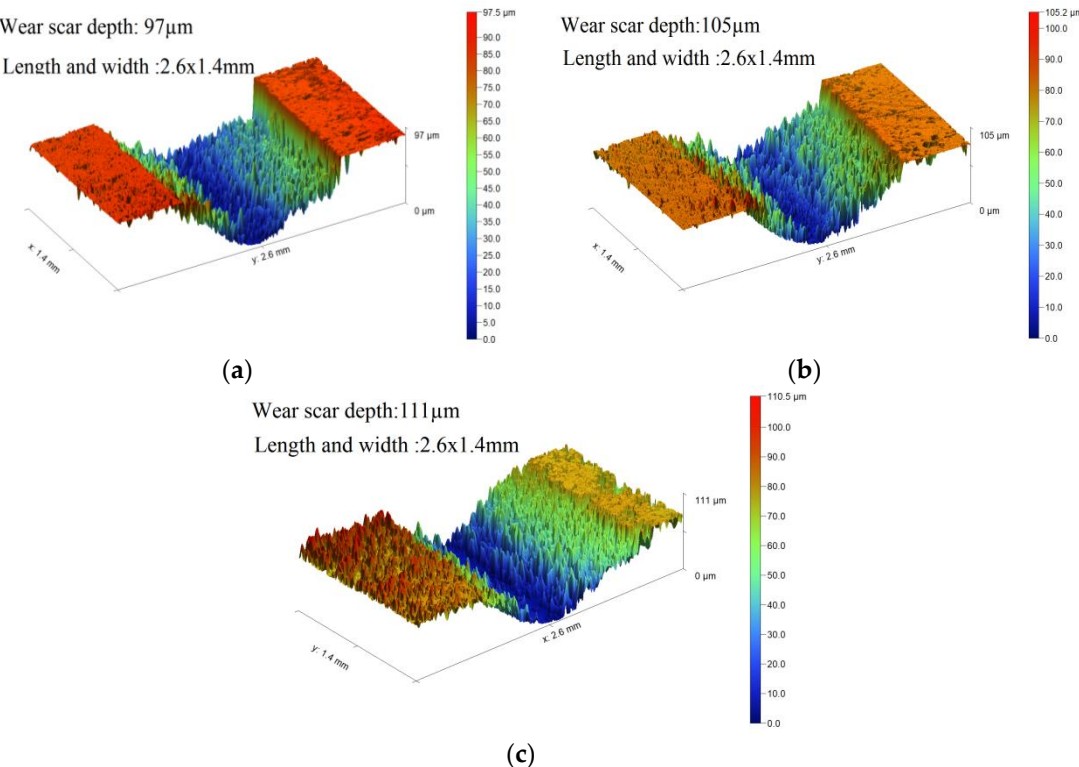

**Figure 9.** 3D morphology of the wear marks under different forces, magnified 10×: (**a**) 20 N; (**b**) 25 N; (**c**) 30 N.

## 4. Conclusions

In this paper, the protective performance of Zn-Al-Mg-TiO$_2$ for marine metal equipment was studied using dynamic salt water corrosion testing, electrochemical testing, and friction and wear testing. Through the analysis of tests results, we have obtained the following conclusions:

- In a corrosive medium, the corrosion rate of Zn-Al-Mg-TiO$_2$ first increases and then decreases, with the Zn-rich phase of the coating consumed first and with a high corrosion rate. After this, the passivation film formed by the corrosion product of Al and Mg covers the coating surface, which can reduce the corrosion rate of the coating.
- TiO$_2$ has excellent photocatalytic self-cleaning performance, the corrosion current density for an ultraviolet radiation experiment is always lower than a no ultraviolet radiation experiment, and the water film attached to the surface of the coating can provide physical shielding for the coating.
- The friction coefficient of the Zn-Al-Mg-TiO$_2$ coating is about 0.35 with a Q235 friction pair. After friction and wear testing, wear marks are relatively smooth, and no bulk material falls off. These results indicate that the Zn-Al-Mg-TiO$_2$ coating has a dense structure, high bonding strength, and great wear resistance.

**Author Contributions:** Conceptualization, S.W.; Methodology, K.W. and S.W.; Validation, K.W., D.W. and W.L.; Data Curation, K.W. and G.W.; Writing—Original Draft Preparation, K.W.; Writing—Review and Editing, S.W. and D.W.; Supervision, T.X. and H.D.; Project Administration, S.W.

**Funding:** This research was funded by the National Natural Science Foundation of China (Nos. 51872122 and 51705199), the Shandong Key Research and Development Plan of China (No. 2016JMRH0218), and Taishan Scholar Engineering Special Funding (2016–2020).

**Conflicts of Interest:** The authors declare no conflict of interest.

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
