# Peer review of "Protective Performance of Zn-Al-Mg-TiO2 Coating Prepared by Cold Spraying on Marine Steel Equipment"

_coatings, doi:10.3390/coatings9050339_

Reviewer 1 Report

The information that TiO2 presents excellent photocatalytic self-cleaning performance and Mg improves the corrosion resistance of Al is not novelty at all, and these are the initial statements submitted in abstract.

In the introduction section authors claim that Zn-Al coatings are focus of current research, yet provide only one reference, which is 6 years old. Reference [4] is not on subject of Zn-Al coatings.

Experimental: what is the equipment used for SEM and EDS? Were the SEM carried out in SE or BSE mode? What was the accelerating voltage? There is a lot of information missing in the Experimental.

What substrate was used for the coatigs?

The manuscript should be language-checked by the native speaker.

Chapter 3.1.

Data presented on Fig.1 is not clear at all! The EDS cannot be deciphered, the Tables are way too small and same are the information on axis X and Y. Furthermore, based on SEM micrographs I assume the analyses were done on point. Why not from larger areas? It would be much better to make the average composition this way.

On Fig 1 I also dont see the discussed difference in dispersion of the aggregates. They are similar to me, maybe authors should have used micrographs with smaller magnification? On the other hand Fig. 1b lacks contrast.

Chapter 3.2.

Authors compare the friction coefficient of the coating and present the results. But its not enough, there is no discussion on the obtained resutls, is it good? Is this difference large? What is the reason behind the difference and what is the mechanism TiO2 affects friction coefficient?

Regarding the micrographs - which phase appears to be lower and which to be higher on Fig 3? The scale is too small and cannot be read. These areas are quite small and it is possible that other areas present higher roughness, thus I am not convinced by authors explanation. What is the average roughness estimated on much larger areas?

Chapter 3.3. 

What were the conditions of the erosion test? The flow of the electrolyte etc? How do authors measured thickness of the corrosion products (they state some layers are thinner)? What is the origin of pits, purely erosive? According to friction test the mechanical properties of the layers are comparable - why do authors observe such huge differences here?

The part of manuscipt devoted to SEM analysis is really difficult to read and should be reworked. 

What is the origin of the cracks visible on Fig. 5d according to authors?

How to authors claim better corrosion resistance based on micrographs? There are corrosion products visible on both of these layers.

There is no discussion regarding Fig. 7

Chapter 3.4

What is the orivin of the potential change at Fig. 8? How many repetitions were done for each sample?

The english quality need to be improved, among many errors what is "reaction speed of coating" ? 

Table 1 and Fig. 9 please provide the information of reference electrode

In Table 1 and 2 - it is hard to compare the current densities directly. It appears that ZnAlMgTiO2 coating is represented by higher current densities at longer exposure times, authors should discuss the differences if they claim any.

Chapter 3.5

Authors havent provide proofs for their suggested mechanism of protection. Furthermore I fail to see significant differences between these two coatings which would validate addition of TiO2. Perhaps authors should use much longer exposure times, since even after 720h the micrographs reveal that both layers provide good separation from the electrolyte.

Author Response

 First of all, I am sorry for my carelessness. I made a lot of changes to my manuscript, added and deleted a lot of content. so I am sorry ,some of your questions  I did not answer

The information that TiO2 presents excellent photocatalytic self-cleaning performance and Mg improves the corrosion resistance of Al is not novelty at all, and these are the initial statements submitted in abstract.

I have revised the statement of TiO2 and Mg in manuscript.

In the introduction section authors claim that Zn-Al coatings are focus of current research, yet provide only one reference, which is 6 years old. Reference [4] is not on subject of Zn-Al coatings.

I have added some new references for this sentence.

Experimental: what is the equipment used for SEM and EDS? Were the SEM carried out in SE or BSE mode? What was the accelerating voltage? There is a lot of information missing in the Experimental.

SEM(JSM-7610F, JEOL, Japan) Imaging in SE mode . The accelerating voltage is 10KV. EDS(JSM-7610F, JEOL, Japan) .

I added the information of experiment I lost

What substrate was used for the coatigs?

The substrate is Q235 (a common carbon structural steel widely used in ships, drilling platforms and other offshore buildings, etc)

Data presented on Fig.1 is not clear at all! The EDS cannot be deciphered, the Tables are way too small and same are the information on axis X and Y. Furthermore, based on SEM micrographs I assume the analyses were done on point. Why not from larger areas? It would be much better to make the average composition this way

using EDS mapping spectrogram to replaced dot spectrogram

Authors compare the friction coefficient of the coating and present the results. But its not enough, there is no discussion on the obtained resutls, is it good? Is this difference large?

Three sets of friction and wear tests were added to the revised manuscript, and analyse the results .

Regarding the micrographs - which phase appears to be lower and which to be higher on Fig 3? The scale is too small and cannot be read. These areas are quite small and it is possible that other areas present higher roughness, thus I am not convinced by authors explanation. What is the average roughness estimated on much larger areas?

Re-tests, a wider range of wear marks was compared. 

What were the conditions of the erosion test? The flow of the electrolyte etc?

In corosion test, the magnetic stirrer was used to stir the corrosive medium, and the corrosion medium is 3.5%NaCl. The speed of magnetic stirre was set to 1000r/min. Before the test, the rest of the sample surfaces was sealed by silicone rubber resin.

What is the origin of the cracks visible on Fig. 5d according to authors?

 From the EDS line scan spectrogram, we find the region is a zinc-rich phase,  but in the hole, the content of Zn is significantly less that indicates zinc was corroded seriously to formed holes

Chapter 3.4(electrochemical test) What is the orivin of the potential change at Fig. 8? How many repetitions were done for each sample?

In electrochemical test, the test time of the open circuit potential was kept above 1 hour, and repeated many times until the potential fluctuation range was less than 1x10-3 v , and the test interval was 24 hours

Table 1 and Fig. 9 please provide the information of reference electrode

The experiment with the traditional three-electrode mode, the counter electrode is a platinum electrode, the reference electrode is a calomel electrode and the working electrode is a 10x10x3mm coating sample

In Table 1 and 2 - it is hard to compare the current densities directly. It appears that ZnAlMgTiO2 coating is represented by higher current densities at longer exposure times, authors should discuss the differences if they claim any.

The table was edited and the results of current density were discussed in revised manuscript

Authors havent provide proofs for their suggested mechanism of protection. Furthermore I fail to see significant differences between these two coatings which would validate addition of TiO2.

In the revised manuscript, I analyzed the protective mechanism of the coating through the test results.

The electrochemical test was carried out under ultraviole and no ultraviole to studied the effect of TiO2on corrosion resistance of coatings, and analyzed the result.

Reviewer 2 Report

The level of English is very, very  poor. The verbs are often missing in long sentences, or they are misplaced. The passive is often not formed by using the past participle. On several occasions prepositions are incorrect or missing too. Several words are misspelled. The syntax of many sentences is erroneous and thus hard to understand. Sentences comparing the quantity use wrong comparatives. Many odd (never seen before) word combinations are used in the manuscript. It is absolutely mandatory to edit the English language and style.

Comments, remarks and questions:

dynamic salt water test – what is the velocity of the water at the sample surface?

Fig. 1:  font size in EDS spectra is too small, it is impossible to read EDS spectra – are they needed at all?       Also fig. 1b – the inset with 0.5 um scale is unreadable.

Is the small difference in friction coefficient (fig. 2) statistically representative?  Same applies for wear scar max. depth (69.89 um vs. 87.38 um) and conclusions drawn from this.

“Compared with friction and wear, the biggest damage of steel components in the marine environment is the electrochemical erosion of the matrix by halogen ions in sea water[18-19].” – “electrochemical erosion” is a term existing only in the machining, not in corrosion.  The ref. 18 and 19 do not even mention the word erosion!

Open circuit potential:  you did not specify to what reference electrode you refer! 

Open circuit potential (OCP) measurement should be at least 1h long. Why is the sampling frequency in fig, 8 so low?

 Fig. 9:  what is the symbol on y axis?  Is this linear or logarithmic scale??  I guess it is logarithmic – then make it clear it is.

Table 1 and 2:  write Icorr with significant digits only

There should be more than one measurement conducted for each line in table 1 and 2. Only then one could then decide about the advantage of Zn-Al-Mg-TiO2 coating compared to Zn-Al.

section 3.5: “the nanostructure of TiO2 blocks the corrosion hole of the coating surface”  -- how can you prove this?  What is your evidence for this claim?

Zn-Al coating does not act only as a barrier but also as a sacrificial anode when the substrate comes into contact with electrolyte (scratch).  It is urgent that this aspect (sacrificial anode) of Zn-Al-Mg-TiO2  is also examined. This is completely missing in this manuscript.

Author Response

First of all, I am sorry for my carelessness. I made a lot of changes to my manuscript, added and deleted a lot of content. so I am sorry ,some of your questions  I did not answer.

dynamic salt water test – what is the velocity of the water at the sample surface?

The speed of magnetic stirre was set to 1000r/min

Fig. 1:  font size in EDS spectra is too small, it is impossible to read EDS spectra – are they needed at all?       Also fig. 1b – the inset with 0.5 um scale is unreadable

A new EDS spectrogram was added, and enlarged the font.

Is the small difference in friction coefficient (fig. 2) statistically representative?  Same applies for wear scar max. depth (69.89 um vs. 87.38 um) and conclusions drawn from this.

For the representativeness of the test, three sets of friction and wear tests were added to the revised manuscript, and analysed the results

“Compared with friction and wear, the biggest damage of steel components in the marine environment is the electrochemical erosion of the matrix by halogen ions in sea water[18-19].” – “electrochemical erosion” is a term existing only in the machining, not in corrosion.  The ref. 18 and 19 do not even mention the word erosion!

You are right. I  revised the statement of "electrochemical erosion". Because I majored in machinery, so I often use this word.

Open circuit potential:  you did not specify to what reference electrode you refer! 

The experiment with the traditional three-electrode mode, the counter electrode is a platinum electrode, the reference electrode is a calomel electrode and the working electrode is a 10x10x3mm coating sample

Open circuit potential (OCP) measurement should be at least 1h long. Why is the sampling frequency in fig, 8 so low?

In electrochemical test, the test time of the open circuit potential was kept above 1 hour, and repeated many times until the potential fluctuation range was less than 1x10-3 v, and the test interval was 24 hours

Figure 9:  what is the symbol on y axis?  Is this linear or logarithmic scale??  I guess it is logarithmic – then make it clear it is.

You are right. I revised the symbol on y axis

Table 1 and 2:  write Icorr with significant digits only

I have revised them in manuscript.

Zn-Al coating does not act only as a barrier but also as a sacrificial anode when the substrate comes into contact with electrolyte (scratch).  It is urgent that this aspect (sacrificial anode) of Zn-Al-Mg-TiO2  is also examined. This is completely missing in this manuscript

I have added this part to the manuscript.

Reviewer 3 Report

The submitted paper concerns the study of Zn-Al and Zn-Al-Mg-TiO2 coatings on steel. After the morphological and chemical characterization, the protective action against corrosion is investigated under salt water, up to 720 h of exposure.

First of all, I suggest revising the English. The manuscript is very difficult to understand.

The context of the research needs to be illustrated more in detail. The citation of other published studies can be helpful.

Explanations about the effects of Mg and nano-TiO2 in isolation should be given. Furthermore, where Mg and nano-TiO2 are used in combination, synergistic effects should be highlighted, if observed. This latter may justify predictable increases in the costs of production.

No information is given about the applied TiO2 nanoparticles. How have been they prepared? Which are their dimensions? Which is their crystallographic form? These data should be added.

Point out the advantaged of using the cold spraying application.

Did the authors evaluate the thickness of the applied coatings?

Some of the cited references deal with photocatalytic activities of TiO2 nanoparticles (e.g., refs 13, 14, 15). Have these properties of TiO2-based materials influence on the protective action of the studied coatings? If so, the effect of light irradiation should be also investigated.

Additional comments:

Line 37: the first author of the listed reference 9 is not Yao; check and correct.

Line 55: give more details on the Q235 substrate.

Section 2.2.: detail how many replicates were performed in each test; in addition, define the detector used for SEM observations.

Line 92 (and line 262): the observations through SEM are not so useful to evaluate how a surface is smooth. Maybe, AFM could be more suitable. Did the authors take into consideration this technique?

Lines-96-97: add a description of the image in the inset box of Figure 1b.

Lines 107-108, Table 1, and Table 2: add deviations to the reported data.

Lines 181-188: revise the captions for Figures 5 and 6, because the letters indicating the images differ from those reported in both captions and text.

Line 191: EDS line-scans, not spectra are reported in Figure 7. I think that such results can be barely reproducible and not provide significant information. In fact, spot EDS analyses (on both the light areas and the lamellar regions) could be appropriate to examine the different components. In this regard, indicate the corrosion time; compare with the results of similar investigations on the Zn-Al-Mg-TiO2 coating.

Line 249: do you mean Figure 10 (a)?

Section 4: the concluding remarks are too much synthetic; I suggest rephrasing.

Lines 317-319 (and line 44): Ref 12 seems not pertinent; please, check.

Author Response

First of all, I am sorry for my carelessness. I made a lot of changes to my manuscript, added and deleted a lot of content. so I am sorry ,some of your questions  I did not answer.

The context of the research needs to be illustrated more in detail. The citation of other published studies can be helpful.

Thanks for your advice. I have added this part content to the manuscript

Point out the advantaged of using the cold spraying application.

I have added this part content to the manuscript

Did the authors evaluate the thickness of the applied coatings?

The thickness of the coating can be characterized by adding the micro-morphology of the cross-section of the coating in the manuscript.

Some of the cited references deal with photocatalytic activities of TiO2 nanoparticles (e.g., refs 13, 14, 15). Have these properties of TiO2-based materials influence on the protective action of the studied coatings? If so, the effect of light irradiation should be also investigated.

The electrochemical test was carried out under ultraviole and no ultraviole to studied the effect of TiO2on corrosion resistance of coatings, and analyzed the result.

Line 37: the first author of the listed reference 9 is not Yao; check and correct

You are right. I  revised  the author name

Lines 107-108, Table 1, and Table 2: add deviations to the reported data

The table was edited and the results  were discussed in revised manuscript

Lines 181-188: revise the captions for Figures 5 and 6, because the letters indicating the images differ from those reported in both captions and text.

You are right. I  revised  it

Line 191: EDS line-scans, not spectra are reported in Figure 7. I think that such results can be barely reproducible and not provide significant information. In fact, spot EDS analyses (on both the light areas and the lamellar regions) could be appropriate to examine the different components. In this regard, indicate the corrosion time; compare with the results of similar investigations on the Zn-Al-Mg-TiO2 coating.

I have revised this section in the manuscript and analyzed the results.

Section 4: the concluding remarks are too much synthetic; I suggest rephrasing.

You are right, Thanks you advices and I re-edited the summary section.

Lines 317-319 (and line 44): Ref 12 seems not pertinent; please, check.

You are right. I  revised  it.

Round  2

Reviewer 1 Report

I fail to see what is the information provided by EDS on Figure 2. True, it does look neat and show the distribution of TiO2 and Zn rich or Mg rich areas, but what is the conclusion drawn on this figure? Authors doesn’t use EDS to compare various materials or to study the degradation of coating immersed in salt water… what is the gain? Much more can be analyzed based on SEM micrographs later in the manuscript. 

Chapter 3.4. The unit for potential is V, not v. When providing it in text or figures one must provide the reference electrode information too (Figures 6 and 7 and in text). What is standard potential of authors reference electrode? 

These plots cannot be called Tafel polarization curves, since Tafel extrapolation does not apply in this case. The process of charge transfer through the coating is diffusion-controlled therefore the Tafel law does not apply. Furthermore, authors have not explained how they assessed Icorr in Table 1. Since Tafel law does not apply in here, one cannot provide strict information but just rough comparison. 

What is the origin of pits, purely erosive? According to friction test the mechanical properties of the layers are comparable - why do authors observe such huge differences here?

There manuscript still must be language-checked. There are some obvious editorial errors in text.

Author Response

I fail to see what is the information provided by EDS on Figure 2. True, it does look neat and show the distribution of TiO2 and Zn rich or Mg rich areas, but what is the conclusion drawn on this figure? Authors doesn’t use EDS to compare various materials or to study the degradation of coating immersed in salt water… what is the gain? Much more can be analyzed based on SEM micrographs later in the manuscript.

 I added some EDS and SEM results analysis

Chapter 3.4. The unit for potential is V, not v. When providing it in text or figures one must provide the reference electrode information too (Figures 6 and 7 and in text). What is standard potential of authors reference electrode?

I have revised unit V and added reference electrode information.
The standard potential of reference electrode  is 0.2801V

These plots cannot be called Tafel polarization curves, since Tafel extrapolation does not apply in this case. The process of charge transfer through the coating is diffusion-controlled therefore the Tafel law does not apply. Furthermore, authors have not explained how they assessed Icorr in Table 1. Since Tafel law does not apply in here, one cannot provide strict information but just rough comparison.

Yes, you're right. The curve is  potentiodynamic polarization curves
The current density in Table 1 is the precise value obtained by fitting the potentiodynamic polarization curves with electrochemical software.

What is the origin of pits, purely erosive? According to friction test the mechanical properties of the layers are comparable - why do authors observe such huge differences here?

Pits on the coating surface are mainly caused by corrosion and Non-uniform spraying . Compared with Fig. 1 (a), there are no obvious holes on the surface. However, after 288 hours of corrosion, there are a lot of holes in Fig. 5. So I think the main reason for the formation of these holes is corrosion. In addition, the friction and wear 3D image is only magnified 10 times, so I think Figure 9 illustrates that the surface of the coating is not smooth enough, not holes. However, maybe you're right. I don't have any evidence that the holes in Fig 5 must have been formed by corrosion, so I revised that statement.

Reviewer 3 Report

The paper has been strongly modified. The quality has been improved, but the manuscript still requires revisions.

Not all the comments have been assessed. Although some of the observations are no longer pertinent, others have not been taken into account.

The following are my remarks:

-        How many samples have been tested? Add this information.

-        Explain if the tests have been performed in replicate.

-        Report the standard deviations of the data (in Table 1, at least).

-        Specify the detector used to perform the SEM observations. Experts in the field are able to identify that a secondary electron detector was used. Conversely, the readers not familiar with this technique cannot understand this experimental detail. On the other hand, only the acronym SE is reported in the caption for Figures 1 and 4.

-        Carefully proofread the manuscript.

Author Response

 1.How many samples have been tested? Add this information

   the dynamic salt water corrosion test with four Samples for different times in each groupelectrochemical test with two samples under light and non light conditions; reciprocate friction wear test was carried out three times for a sample in different force.

2.Explain if the tests have been performed in replicate

  each experiments were repeated twice, taking the most representative group for the study

3.Report the standard deviations of the data (in Table 1, at least).

   I have added it to Table 1

4.Specify the detector used to perform the SEM observations. Experts in the field are able to identify that a secondary electron detector was used. Conversely, the readers not familiar with this technique cannot understand this experimental detail. On the other hand, only the acronym SE is reported in the caption for Figures 1 and 4.

  I explained it in 115-116

Round  3

Reviewer 1 Report

I accept authors corrections in all aspects but one. Sadly, this one is the most critical, hence I cannot advice for publication of the manuscript in its current form.

In the previous revision I have pointed out that the Tafel law does not apply in the studied case, since the studied process is diffusion controlled one. Authors agreed with my criticism and changed names of plots. However, they are recklessly stating that the Icorr values are precise and obtained via electrochemical software. Let me ask thauthors - by what means do the software evaluate the polarization plots? The software use Tafel extrapolation method which may be applied only if Tafel law is met. And this is not such case! Therefore one cannot provide precise information about corrosion currents, but merely compare the plots.

Author Response

First of all, I apologize for my recklessness. By consulting information and your criticism, I learned that the studied process is diffusion controlled one, and the corrosion rate is controlled by diffusion process. At this time, the Tafel extrapolation method can not be measured to test corrosion current, and it is impossible to obtain accurate corrosion current density. Only obtain a approximation by comparison. Therefore, I revised the content of the manuscript.